# Primary Membranous Glomerulonephritis: The Role of Serum and Urine Biomarkers in Patient Management

**DOI:** 10.3390/biomedicines7040086

**Published:** 2019-11-01

**Authors:** Sadiq Mu’azu Maifata, Rafidah Hod, Fadhlina Zakaria, Fauzah Abd Ghani

**Affiliations:** 1Histopathology Unit, Department of Pathology, Faculty of Medicine and Health Science, Universiti Putra Malaysia, Serdang, Selangor 43400, Malaysia; teemoh87@gmail.com; 2Physiology Unit, Department of Anatomy, Faculty of Medicine and Health Science, Universiti Putra Malaysia, Serdang, Selangor 43400, Malaysia; rafidahhod@upm.edu.my; 3Department of Physiology, Faculty of Basic Medical Science, College of Medicine, Federal University Lafia, Lafia, Nasarawa 950102, Nigeria; 4Nephrology Unit, Department of Medicine, Faculty of Medicine and Health Science, Universiti Putra Malaysia, Serdang, Selangor 43400, Malaysia; n_fadhlina@upm.edu.my

**Keywords:** M-type phospholipase A2, thrombospondin type containing domain A7, retinol binding protein, beta-2 microglobulin, membranous glomerulonephritis, neutral endopeptidase

## Abstract

The detection of phospholipase A2 receptor (PLA_2_R) and thrombospondin domain containing 7A THSD7A among primary membranous glomerulonephritis (MGN) patients transformed the diagnosis, treatment monitoring, and prognosis. Anti-PLA_2_R can be detected in 70–90% of primary MGN patients while anti-THSD7A in 2–3% of anti-PLA_2_R negative primary MGN patients depending on the technique used. Serum and urine samples are less invasive and non-invasive, respectively, and thus can detect the presence of anti-PLA_2_R and anti-THSD7A with higher sensitivity and specificity, which is significant in patient monitoring and prognosis. It is better than exposing patients to a frequent biopsy, which is an invasive procedure. Different techniques of detection of PLA_2_R and THSD7A in patients’ urine and sera were reviewed to provide newer and alternative techniques. We proposed the use of biomarkers (PLA_2_R and THSD7A) in the diagnosis, treatment decision, and follow-up of patients with primary MGN. In addition, other prognostic renal biomarkers like retinol binding protein (RBP) and beta-2 microglobulin were reviewed to detect the progression of renal damage for early intervention.

## 1. Introduction

Membranous glomerulonephritis (MGN) is characterized by the deposit of the immune complex at the subepithelial region of the glomerular membrane and the formation of perpendicular projection similar to the glomerular basement membrane (GBM) found between podocyte cytoplasm and GBM [1]. The histological characteristics of MGN include GBM thickening, staining of the C3 complement along the periphery of glomerular capillaries, granular staining for immunoglobulin G subtype, and deposition of the immune complex at the subepithelium found exclusively in primary MGN [2].

Studies have shown that MGN is the most common cause of nephrotic syndrome among adults, consisting of up to 20% of cases among Hispanic and African Americans. Whites are the most affected, followed by Asians, Africans, and Hispanics [3,4]. MGN accounts for 1.5–9% of the nephrotic syndrome among children and 21–35% in adults. The disease is predominantly seen among adults aged 40–50 years, with a male to female ratio of (2:1)–(3:1) [5].

Idiopathic or primary MGN is the commonest type seen in about 75% of the patients with MGN while the remaining 25% manifest as a secondary disease to other conditions, mainly infection [6,7,8,9]. 

It is very difficult to differentiate between primary and secondary MGN based on their clinical presentations and laboratory evaluations. Therefore, good leading history, clinical, and laboratory findings (including renal biopsy) are crucial in differentiating the two types of MGN [10]. Diagnosis of MGN is traditionally made through electron microscopy, light microscopy, and immunofluorescence techniques from renal biopsy [11].

There are a lot of attempts to use less invasive methods, like serum samples, or non-invasive methods, like urine samples, to make a diagnosis and monitor MGN patients immediately following the detection of anti-phospholipase A2 receptor (PLA_2_R) and anti-thrombospondin domain containing 7A (THSD7A) in the serum of primary MGN patients. Despite these tremendous achievements, making a diagnosis using serum alone is still difficult due to the lack of a cut-off point value universally agreed on for diagnosis and follow-up. Moreover, certain forms of secondary diseases are associated with these biomarkers.

Hence, this review aims at highlighting the newer techniques used in making a diagnosis of MGN by detecting biomarkers in serum and urine samples of MGN and the importance of those biomarkers in patient management.

## 2. Pathogenesis

Immune complex formation in the subepithelial surface of the glomerular complex membrane is central to the formation of membranous glomerulonephritis [12]. Three major mechanisms have been proposed so far [1,2].

The first hypothesis emphasizes on passive entrapment of preformed immune complex due to higher intraglomerular pressure and negatively charged capillary which force the protein across the glomerular capillary wall. An example is in lupus nephritis [13,14].

The second hypothesis involves the localization of the circulating antigens in the subepithelial aspect of the glomerular basement membrane, as such, the antigen forms in-situ immune complex deposit with the antibodies. This is seen in hepatitis B [15,16], hepatitis C [17,18], Helicobacter pylori [19], and patients diagnosed with MGN [20,21].

The third mechanism focuses on the binding of autoantibodies to antigens, as well as to podocyte membrane, leading to immune complex deposition. Numerous studies were conducted to describe the pathogenesis of MGN, ranging from the Heymann rat model of 1959 where membranous glomerulonephritis was induced using Lewis rats with crude kidney extract [22]. The use of megalin in the 1980s by Kerjaschki and Farquhar showed oxygen radicals responsible for glomerular damage resulting in proteinuria [23]. No anti-megalin antibody was recorded to have been found in patients with MGN. Therefore, megalin cannot be associated with proteinuria in human [12,24].

## 3. Materials and Methods

Pubmed, Google Scholar, Springer, and Science Direct were searched for manuscripts published in English using keywords, such as “*primary OR idiopathic membranous nephropathy*” alone or in combination with “*prognosis*” or “*clinical features*”. Review articles were cited to provide more details and additional references.

### 3.1. The Biomarkers

Neutral endopeptidase (NEP) is the first human biomarker for MGN identified in early 2000s and is seen in patients with alloimmune neonatal MGN. This involves vertical transmission from a mother to her offspring following sensitization during previous pregnancy [25]. Therefore, it is very important to screen families with a history of membranous nephropathy as part of their antenatal care [25,26].

Beck et al. (2009) conducted a breakthrough study where a M-type phospholipase A2 receptor (PLA_2_R) was detected as a biomarker for MGN using Western blot technique. More recently, another biomarker, thrombospondin domain containing 7A (THSD7A) was discovered to complement PLA_2_R. This biomarker is detected in seronegative anti-PLA_2_R primary MGN patients except for some rare conditions where anti-PLA_2_R and anti-THSD7A can be detected [27,28]. Anti-THSD7A cannot be detected in a normal individual or patients with secondary MGN [29,30].

Anti-THSD7A occur in 2.5–5% of patients with idiopathic MGN and does not appear in secondary MGN. Except for few where both PLA_2_R and THSD7A were found positive, THSD7A is only detected among those MGN patients who are PLA_2_R negative [31,32].

#### 3.1.1. Clinical Feature

Nephrotic range proteinuria is the commonest presentation among MGN patients. It occurs in 70–80% of patients associated with edema, hypoalbuminemia, and hyperlipidemia, while the remaining patients present with subnephrotic range proteinuria [3,4,33,34,35,36]. The renal function may be normal or slightly impaired at diagnosis. An abrupt change in renal functions may call for a prompt investigation to look for possible superimposed conditions like bilateral renal thrombosis, drug toxicity, and crescentic glomerulonephritis [37]. Other features include hematuria; hypertension is mostly not specific to idiopathic membranous nephropathy [38].

A study involved the administration of THSD7A-specific IgG to mice, thereby leading to massive proteinuria and histomorphological changes of MGN. The above findings showed that anti-THSD7A antibodies might interfere with the integrity of podocyte resulting in damage of cells and subsequently proteinuria [39].

Most of the patients presenting with subnephrotic proteinuria are asymptomatic and have a natural history different from those with nephrotic range proteinuria. About 40% will have spontaneous remission, needing just conservative management while the remaining 60% may develop nephrotic range proteinuria within 2 years of presentation, especially when the anti-PLA_2_R antibody is still present [40,41]. The progression of the disease is four times accelerated, which becomes synonymous to those that presented with nephrotic syndrome ab initio [42]. This is another scenario in which anti-PLA_2_R measurement may be important [5].

#### 3.1.2. PLA_2_R-Related Sarcoidosis and Hepatitis B Virus (HBV) Infection

Sarcoidosis is rarely seen in glomerular diseases. However, when it occurs, it is frequently associated with MGN [43,44]. The pathogenesis linking MGN to sarcoidosis remains unclear due to the inability to identify a target antigen or specific antibody linking the two diseases. Notably, while MGN resulted from an autoimmune reaction involving type-2 helper (Th2) cells, sarcoidosis is associated with type-1 helper (Th1) cells [45,46]. Stehle et al. (2015) demonstrated a high prevalence of anti-PLA_2_R antibodies in the serum of MGN patients with sarcoidosis [44].

In addition to sarcoidosis, PLA_2_R was detected in 64% hepatitis B virus (HBV) related MGN [47]. This may be due to co-localization of PLA_2_R and HB_s_Ag in HBV-related MGN due to chronic HBV infection [48,49].

#### 3.1.3. THSD7A and Malignancy

Patients with anti-THSD7A were found to be at risk of reoccurrence after transplant [50]. It has been observed that the messenger RNA for THSD7A was detected in gall bladder carcinoma and not in the normal gall bladder [51]. Serum samples of 1276 patients with MGN were screened for anti-THSD7A. Forty of them tested positive for anti-THSD7A and 8 out of the 40 developed malignancy within an average follow-up period of 3 months. Thus, this finding denoted that THSD7A is associated with malignancy [52]. In another study involving 81 patients with breast cancer and 20 with colorectal cancer, THS7A expression was detected by immunohistochemical stains in 100% and 97.5% of breast cancer and colorectal cancer, respectively [53].

#### 3.1.4. Role of Biomarkers in Kidney Transplant

The discovery of PLA_2_R does not only help in differentiating primary and secondary MGN but also helps in pre- and post-kidney transplantation by identifying those at risk of post-transplantation failure for intensive therapy and monitoring [54,55,56]. About 30–50% of primary MGN patients are at risk of reoccurrence following kidney transplant especially among those with very high anti-PLA_2_R antibodies titer [57,58]. Anti-PLA_2_R titer should be considered post-transplantation, even though the threshold of anti-PLA_2_R level determining the reoccurrence is not clear [59] (Figure 1).

### 3.2. Detection of Biomarkers

The following methods can be used in detecting PLA_2_R and THSD7A antibodies in the patients’ sera: Enzyme-Linked Immunosorbent Assay (ELISA) method, Laser Bead Immunoassay (ALBIA), Luciferase Immunoprecipitation System (LIPS) [60]. Recently, different methods of detecting anti-PLA_2_R and THSD7A are available commercially due to the rise in their clinical applications in patient management.

#### 3.2.1. Western Blot Technique

Western blot technique is the first method used to detect the expression of anti-PLA_2_R in the patient’s serum with primary MGN [61]. In the Western blot technique, proteins are separated based on their molecular weight through gel electrophoresis [62]. The technique was first reported to have a sensitivity of about 70% with up to 96% specificity [61]. There were also reports on an improved variant of the Western blot technique (specially designed to detect very low anti-PLA_2_R) with sensitivity even greater than 90% [63]. However, the method is only suitable for a smaller sample size and also requires expertise [64].

#### 3.2.2. Recombinant Cell-Indirect Immunoassay (RC-IFA)

In this case, the substrate is human cell-line HEK293-overexpressing the full-length PLA_2_R_1_ [65]. RC-IFA can be used to make a diagnosis and monitor primary MGN patients. It was shown to have a sensitivity of 75% and a specificity of nearly 100% [66]. RC-IFA has been considered the most suitable method for the detection of anti-PLA_2_R1 [67]. It can also be used as a reference technique to determine several antibodies against the membrane proteins of the outer surface like NMDR in autoimmune encephalitis [68], aquaporin 4 in neuromyelitis optica [69], and Crohn’s disease [70].

Despite all the advantages mentioned above, it has its limitations. RC-IFA requires technical know-how and special equipment like fluorescence microscopy. It is liable to subjective interpretation and it can only give a semi-quantitative variable.

#### 3.2.3. Time-Resolved Fluoroimmunoassay (TRFIA)

This is a highly specific antigen–antibody binding reaction by measuring light-emission from labels conjugated from protein [71,72]. This assay has high sensitivity, used in the quantitative measurement of serum PLA_2_R-Ab immunoglobulin (IgG4) [73].

#### 3.2.4. Laser Bead Immunoassay (ALBIA)

Behnert et al. (2013) reported the use of Laser Bead Immunoassay (ALBIA) using an in vivo expressed recombinant human PLA_2_R to take care of limitations of RC-IFA (quantitative assay, high objectivity in assessment) [74]. Further study was carried out to compare with two other commercially available immunoassays. It was proved that ALBIA correlates better with RC-IFA than with ELISA (*p* = 0.003) and the overall result showed sensitivity and specificity of 60%/98.6%, and 56.2%/100% for ALBIA and RC-IFA, respectively [75]. Despite this remarkable result for ALBIA, it is not available commercially.

#### 3.2.5. Luciferase Immunoprecipitation System (LIPS)

Another technique is the LIPS assay that makes use of light-emitting proteins. This can detect different types of antibodies, including anti-PLA_2_R [76]. The PLA_2_R LIPS assay is quantitative and highly sensitive. It has a sensitivity of nearly 100% and a specificity of 100% and is better than most of these methods of detecting PLA_2_R. It can also positively correlate with proteinuria and disease process (*p* < 0.005) [60]. More studies are needed to prove the above claim, and its uses are limited to research only (not yet available commercially).

#### 3.2.6. Enzyme-Linked Immunosorbent Assay (ELISA)

There is an urgent need to develop a standardized ELISA to overcome the above-mentioned shortcomings and to give identical diagnostic accuracy for better clinical importance. This involves the expression of PLA_2_R1 in HEK293. This technique was used to analyze sera from 200 primary MGN patients, 27 secondary MGN, and 291 healthy individuals. The results indicated a remarkable sensitivity of 78% and a specificity of 91%. The result has correlated significantly well with clinical findings of patients and the results obtained from RC-IFA [64].

A comparative study involving different methods of detecting PLA_2_R antibody among 158 patients was conducted of which 142 were primary and 16 were secondary MGN. Western blot, ELISA, and IIFT techniques were compared, and the results showed a specificity of 97% for all techniques, a sensitivity of 68% for IIFT, and 72% for both ELISA and the Western blot technique. The ELISA technique may be the preferred method because it can be used for a larger sample size, both qualitative and quantitative measurements. It is less time consuming, requires less technical know-how, and can be interpreted objectively. This clearly showed the superiority of the ELISA method in terms of commercial availability and clinical application [77].

Western blot, ELISA, and RC-IFA are widely used due to their commercial availability and technically. The ELISA technique is widely used compared to other methods due to its ability to measure both qualitative and quantitative assays, and also for its affordability. Table 1 below shows the superiority of the ELISA technique over other methods.

#### 3.2.7. Detection of Anti-PLA_2_R and Anti-THSD7A in Serum

A meta-analysis involving 19 studies and 1160 patients was conducted to investigate the clinical importance and the accuracy of histological and serological PLA_2_R in differentiating primary and secondary MGN. The overall results showed a sensitivity of 0.68, specificity of 0.97, and the diagnostic odds ratio (DOR) was 3.75, while the area under the receiver operating curve (AUROC) was 0.82 for serum anti-PLA_2_R with a substantial heterogenicity (I^2^ = 86.42%). In the case of PLA_2_R staining, the overall sensitivity was 0.78, specificity was 0.91, and DOR (34.70) and AUROC (0.84) without heterogenicity (I^2^ = 0%). Thus, serum anti-PLA_2_R can be used to make a diagnosis of primary MGN, but in the case of serum anti-PLA_2_R negative patients, clinical presentation and biopsy must be considered before making the diagnosis [78,79].

Another study involved 57 biopsy-proven primary MGN, 62 non-MGN, and 22 healthy individuals using the ELISA technique to quantify anti-PLA_2_R in the serum. The results obtained show that at a cut-off value of 2.0 RU/mL, the sensitivity and specificity were 75% and 82.5%, respectively. At 2.6 RU/mL, the sensitivity and specificity were 78.9% and 91.7%, respectively, while the sensitivity and specificity at 20 RU/mL were found to be 50.9% and 96.4%, respectively. As the cut-off value increases, the sensitivity and specificity also increase [80].

THSD7A is a form of membrane-associated *N*-glycoprotein found within endothelial cells of the human umbilical vein and also expressed in blood vesicles of the placenta [81].

A study conducted in 154 primary MGN patients demonstrated that 15 of the 154 patients have antibodies for only anti-THSD7A [58]. Subsequently, more researches were conducted to detect the level of anti-THSD7A in the serum of MGN patients. Anti-THSD7A can be detected in serum and tissue of primary MGN patients with THSD7A [82]. Unlike PLA_2_R, THSD7A does not show any significant preference for serum creatinine, albumin, and proteinuria levels [83]. No significant correlation was found between THSD7A and proteinuria [52].

In a meta-analysis of 10 different studies involving 4121 patients with primary MGN and based on sample size, race, and detection method, it was found that anti-THSD7A was 3% in all patients and 10% among those without anti-PLA_2_R. Thus, this clearly showed a significant difference in the prevalence of anti-THSD7A based on the sample size but not many differences were observed among the races [39].

In a study conducted on 1318 biopsy-proven primary MGN, 31 stained positives for THS7A showed a strong correlation with IIFT results (*p* < 0.001) [82].

#### 3.2.8. Detection of Anti-PLA_2_R and Anti-THSD7A in Urine

A urine sample is non-invasive and can detect renal damage more than serum. Therefore, it is important to demonstrate whether anti-PLA_2_R can be detected in urine. To do this, a study was conducted on 28 primary MGN and 12 secondary MGN patients in China using ELISA and IIFT. The result showed that 18 of the 28 (64.3%) primary MGN patients tested positive for IIFT serum PLA_2_R, while 19 of the 28 (67.9%) had IIFT positive urinary anti-PLA_2_R. The antibody titer of anti-PLA_2_R from primary MGN patients in urine and serum is higher than the corresponding titers from secondary MGN (*p* < 0.05). Statistical analysis indicated a positive correlation between urinary anti-PLA_2_R and serum anti-PLA_2_R. More studies needed to prove that anti-PLA_2_R can be detected in the urine of primary MGN patients [84].

Despite several studies involved in the detection of THSD7A in tissue and serum, no known published study is regarding its detection in the patients’ urine.

### 3.3. Diagnosis

Previous studies showed that anti-PLA_2_R is now an established parameter for diagnosing primary MGN, differentiating it from secondary type, monitoring treatment, and prognosis [85]. The antibody titer helps in monitoring treatment more than proteinuria as the change in titer is immunological, so it precedes the change in proteinuria [11].

All patients with biopsy-proven MGN should be screened for anti-PLA_2_R/THSD7A, as well as hepatitis C, hepatitis B, lupus nephritis antigens, and malignancies to rule out secondary causes [86,87,88].

Most ELISA authors define positivity of anti-PLA_2_R using a cut-off point of 20 RU/mL, some use 14 RU/mL, 2.6 RU/mL, or 2 RU/mL as their cut-off point value to define positivity [89,90,91,92]. In some cases, the cut-off point value is obtained by measuring the anti-PLA_2_R of apparently normal subjects without any renal compromised [93].

Though detection of anti-PLA_2_R and anti-THSD7A in serum were found to be reliable, biopsy remains the best option in the diagnosis of primary MGN. A study conducted on 42 biopsies has proven primary MGN with serum samples collected at the time of biopsy. The resultant sensitivities and specificities were 0.74 and 0.57 for renal glomerular deposit and circulating anti-PLA_2_R in serum, respectively, with 3 patients who had no PLA_2_R but detectable anti-PLA_2_R in their serum. Furthermore, 18 patients who were serum anti-PLA_2_R negative have a glomerular deposit of PLA_2_R. This study suggested that both biopsy and serum are very important in making the diagnosis of primary MGN [94].

### 3.4. Treatment of Idiopathic MGN and Further Therapy

Serum and urine biomarkers (PLA_2_R and THSD7A) are now used in monitoring the efficiency of immunosuppressive therapy. The biomarkers can also be used to compare two immunosuppressive drugs by measuring the serum level of PLA_2_R and THSD7A before, during, and after treatment [95,96,97,98]. Rituximab can be used to reduce PLA_2_R. However, the total dose needed to clear anti-PLA_2_R remains unclear and may be patient dependent [99].

Another suggestion was the use of drugs that inhibit the factors that activate autoreactive B cells. In this case, Belimumab acts by reducing the production of autoantibodies by binding to a B-lymphocyte Stimulator (BLyS) [100]. In a study involving 14 patients, it was found that Belimumab caused significant reduction in anti-PLA_2_R and proteinuria and normalized the serum albumin level [101].

### 3.5. Prognosis

Recent studies conducted within 5 years have shown that anti-PLA_2_R concentration is correlated with urinary protein and disease activities; the antibodies are usually undetectable in spontaneous or drug-induced remission patients and reappear when there is relapse [102,103,104,105,106].

Toronto risk score has been used to predict the prognosis of MGN patients. It is calculated based on creatinine clearance at diagnosis, highest sustained 6 months period of proteinuria, and slope of creatinine clearance over 6 months with an accuracy level of up to 90%. However, there are challenges associated with this method which include complex calculation and prolonged observation of up to 18 months which may delay treatment [107]. Recently, biomarkers like PLA_2_R, retinol binding protein (RBP), and beta-2 microglobulin can be used to predict prognosis among patients with MGN [108].

RBP is considered to be a prognostic biomarker for MGN and other chronic kidney diseases. The high value of RBP is associated with poor prognosis [109,110].

β2-microglobulin can predict the prognosis of MGN [111,112]. β2-microglobulin has 88% sensitivity and 91% specificity in determining the prognosis in renal failure with the threshold level at 40 mg/min [111]. However, when re-evaluated, both biomarkers show low sensitivity and specificity compared to the initial result. This may be due to conservative therapy. There are no significant differences between prognostic accuracies from β2-microglobulin and Toronto risk score [107].

A retrospective study involving 33 non-nephrotic MGN patients showed that anti-PLA_2_R positive patients (48%) were more at risk of progressing towards a nephrotic syndrome compared to anti-PLA_2_R negative patients. In addition, patients with high anti-PLA_2_R titer are more at risk of adverse effects of immunosuppressive drugs and end-stage renal disease compared to those with no anti-PLA_2_R by the end of the follow-up [105]. More studies involving a large number of patients are needed to confirm the above claim, as the small number of patients in this study and the too short follow-up duration rendered it hard to categorically determine the outcome.

To accurately predict the progression of MGN, the watchful waiting method was adopted. This involves 24 h urinary protein and creatinine clearance monitoring for at least 6 months and comparing the result with nephrotic range proteinuria [108].

It is important to know that the presenting proteinuria is inversely proportional to the rate of spontaneous remission [112]. Further, it was observed that there is a high chance of spontaneous remission if a 50% reduction in proteinuria is achieved within the first year irrespective of the initial level of proteinuria. About 32% can undergo spontaneous remission within 14 months and up to 62% in 5 years especially among MGN patients with decreased (low) anti-PLA_2_R and anti-THSD7A [29,41,113].

More recent studies support a watchful waiting approach and also indicated that spontaneous remission can occur even when the presenting proteinuria is greater than 12 g/day [113].

Further studies involving natural history and spontaneous remission in Southeast Asia and Malaysia are needed.

## 4. Conclusions

The standardized ELISA method is the best so far considering its ability to measure both qualitative and quantitative variables, it requires less time, it is easier to perform, it has high sensitivity and specificity, and is also readily available and affordable commercially.

Serum or urine samples should be used to determine the level of anti-PLA_2_R/anti-THSD7A to make a diagnosis, monitor patients’ treatment, and to determine the prognosis especially among patients who cannot withstand renal biopsy. Serum and urine samples can determine when to commence or stop immunosuppressive therapy.

## Figures and Tables

**Figure 1 biomedicines-07-00086-f001:**
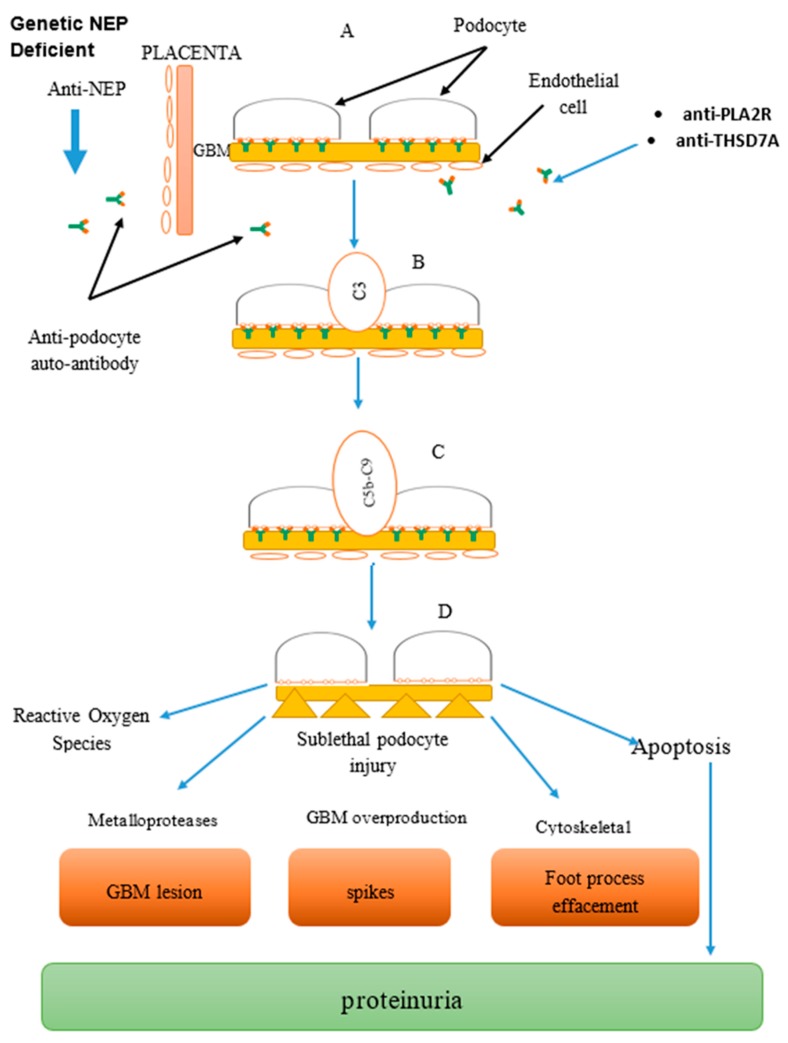
Combined image: Mechanism of anti-podocyte (anti-neutral endopeptidase (NEP), anti-phospholipase A2 receptor (PLA_2_R), anti-thrombospondin domain containing 7A (THSD7A)) antibody-mediated disease in membranous glomerulonephritis (MGN), part of glomerular basement membrane (GBM). Formation of complexes: (**A**) Antigen–antibody reacts to form complexes at the podocyte. (**B**) Complement activation system via the classical and alternative pathway. (**C**) Formation of complement membrane attack complex leading to cell injury. (**D**) Podocyte injury leading to proteinuria and cell death.

**Table 1 biomedicines-07-00086-t001:** Showing various techniques used in detecting PLA_2_R antibody.

Techniques	No. of Subjects	Sensitivity (%)	Specificity (%)	References
Western blot	37	70	96	[61]
ALBIA	157	60	96	[75]
LIPS	45	97	100	[60]
ELISA	200	96.5	99.99	[64]
TRFIA	39	89.7	100	[73]
RC-IFA	75	75	100	[66]

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
