# Peer review of "Primary Membranous Glomerulonephritis: The Role of Serum and Urine Biomarkers in Patient Management"

_biomedicines, 2019, doi:10.3390/biomedicines7040086_

Round 1

Reviewer 1 Report

     Authors have compiled many papers and discussed the role of serum/urine biomarkers in the management of membranous nephropathy. The intention is good, but there are issues having be taken care of.

In table 1, page 8, there is a statement "...despite the fact that RC-IFA is considered more specific and sensitive...". Numbers in Table 1 clearly does not support such a statement, and authors recommended ELISA as the methods of choice in the measurement. Please clarify this. Figure 2 describe the method of ELISA. It is clear. However, I wonder the necessity of this figure. Is there any difference of the content in this figure with others? There are many attempts trying to utilize less invasive ways to replace the role of invasive procedures in the diagnosis of diseases. So far, we have not seen hardcore success in this perspective. Can authors address your opinions about this in the text? Authors mentioned that there are some MN patients who might have spontaneous remission in different perioid of time, as in page 10. Do authors know any information about the use of these biomarkers in these patients with spontaneous remission? I think it would be the best if we can identify patients who are going to have high possibility of spontaneous remission, before the use of steroids or other immunosuppressant therapy. I think that 2.5.2 and 2.5.3 should not be in other place of text. Please re-organize.

Author Response

Dear reviewer,

Thank you for your time, observations and suggestions.

kindly find attached response below.

Thank you. 

1.In table 1, page 8, there is a statement "...despite the fact that RC-IFA is considered more specific and sensitive...". Numbers in Table 1 clearly does not support such a statement, and authors recommended ELISA as the methods of choice in the measurement. Please clarify this.

Response: Thank you for your observation. It was indeed an error and it has been noted and corrected.

Most of the reported studies demonstrated that RC-IFA is more specific and sensitive compared to other methods.

2. Figure 2 describe the method of ELISA. It is clear. However, I wonder the necessity of this figure. Is there any difference of the content in this figure with others?

Response: Thank you for your observation, the figure has been removed.

3. There are many attempts trying to utilize less invasive ways to replace the role of invasive procedures in the diagnosis of diseases. So far, we have not seen hardcore success in this perspective. Can authors address your opinions about this in the text? Authors mentioned that there are some MN patients who might have spontaneous remission in different perioid of time, as in page 10. Do authors know any information about the use of these biomarkers in these patients with spontaneous remission? I think it would be the best if we can identify patients who are going to have high possibility of spontaneous remission, before the use of steroids or other immunosuppressant therapy.

Response: Thank you for your observation, we are currently working on the detection of biomarkers in  urine and serum to reduce the rate of biopsy and improve 

patients prognosis and treatment. It has been addressed in the text.

It is important to identify those patients that are likely to have spontaneous remission earlier to avoid or reduce the use of steroids or immunosuppressive therapy on the patients. So far, to the best of our knowledge, those that may likely have spontaneous remission usually have lesser anti-PLA2R titre compare with those that may require immunosuppressive therapy. So those patients can benefit from watchful waiting method. Likewise, they may have > 50% reduction in proteinuria within the first one year. Therefore anti-PLA2R can play significant role in detecting them.

4.I think that 2.5.2 and 2.5.3 should not be in other place of text. Please re-organize.

Response: Thank you for your suggestion, it has been reorganized and merged into prognosis section.

Reviewer 2 Report

The paper by Maifata et al gives an overview of the developments in the field of primary membranous glomerulonephritis. I think it’s a nice overview paper but it needs to be revised significantly. Below I raise a couple of points that should be addressed:

Major
- the manuscript needs to be revised by a native English speaker and writer; some universities have departments for this.
- the references need to be cited Vancouver style, numerically, as is custom in the medical literature. The references also need to be revised, doi numbers are not necessary if the article has appeared in print with a year, volume nr and page nrs.
- It is unclear how the authors performed their search. I think a search strategy should be included with MeSH and free text terms.
- I don’t understand how 2.2.1 pathogenesis is part of the material and methods section. Pathogenesis should be a separate section.
- figure 1: the figure should be revised, it can be made more compact; it is clear it was made in power point or word but the authors could at least have removed the spelling check which is still in the print screen
- figure 2 can be removed, it has no place in this manuscript
- section 2.4 “clinical feature(s)” can be moved to 2.2 since it gives a sort of introduction about primary MGN.
- section 2.5.1 “prognosis” should be the last section in the manuscript. The last sentence of this section nicely wraps up the paper in that case.
- section 2.6 and 2.7 can be merged into one section

Minor
- page 1, line 37: “granule of immunoglobulin G” should be granular staining for IgG
- page 2, line 3: “commonest” should be “most common”
- page 2, line 38: this sentence should be rephrased, it is not easily comprehensible
- page 3, line 8: this sentence should be rephrased
- page 7, line 5: rephrase this sentence
- page 8, line 17: rephrase, this sentence is not comprehensible
- page 8, line 24: the abbreviation of THSD7A has already been used so does not need to be written out again.

Author Response

Dear Reviewer,

Thank you for your time, observations and suggestions.

kindly find attached response below.

Thank you. 

Major
- the manuscript needs to be revised by a native English speaker and writer; some universities have departments for this.

response: The manuscript has been extensively revised by 2 different individual who are experts in English language.

- the references need to be cited Vancouver style, numerically, as is custom in the medical literature. The references also need to be revised, doi numbers are not necessary if the article has appeared in print with a year, volume nr and page nrs.

response:References has been changed to Vancouver style as you suggested and were thoroughly revised - It is unclear how the authors performed their search. I think a search strategy should be included with MeSH and free text terms.

response:A search strategy has been included together with keywords used in the searching

- I don’t understand how 2.2.1 pathogenesis is part of the material and methods section. Pathogenesis should be a separate section.

response:Pathogenesis section has been separated from the material and method section. It is now independent

- figure 1: the figure should be revised, it can be made more compact; it is clear it was made in power point or word but the authors could at least have removed the spelling check which is still in the print screen

- figure 2 can be removed, it has no place in this manuscript

response:The figure has been edited and the spelling check was removed.

- section 2.4 “clinical feature(s)” can be moved to 2.2 since it gives a sort of introduction about primary MGN.

response: Clinical feature section has been moved up to be part of introduction to biomarkers

- section 2.5.1 “prognosis” should be the last section in the manuscript. The last sentence of this section nicely wraps up the paper in that case.

response:Prognosis section, become the last section as suggested. 

- section 2.6 and 2.7 can be merged into one section

response:Section 2.6 and 2.7 have been merged into one section 

Minor
- page 1, line 37: “granule of immunoglobulin G” should be granular staining for IgG

response:Noted and corrected. Thank you.

- page 2, line 3: “commonest” should be “most common”

response:Corrected.

- page 2, line 38: this sentence should be rephrased, it is not easily comprehensible

response:Sentence has been rephrased. Thank you for your comment.

- page 3, line 8: this sentence should be rephrased

- page 7, line 5: rephrase this sentence - page 8, line 17: rephrase, this sentence is not comprehensible

response: Sentences have been rephrased.

- page 8, line 24: the abbreviation of THSD7A has already been used so does not need to be written out again.

response:Noted and corrected

Round 2

Reviewer 2 Report

The authors have adequately replied to my questions and have improved the manuscript.